# Pregnancy Outcomes in Women with Biventricular Circulation and a Systemic Right Ventricle: A Systematic Review

**DOI:** 10.3390/jcm13237281

**Published:** 2024-11-29

**Authors:** Triantafyllia Grantza, Alexandra Arvanitaki, Amalia Baroutidou, Ioannis Tsakiridis, Apostolos Mamopoulos, Andreas Giannopoulos, Antonios Ziakas, George Giannakoulas

**Affiliations:** 1First Department of Cardiology, AHEPA University General Hospital, School of Medicine, Aristotle University of Thessaloniki, 54636 Thessaloniki, Greece; bamalia27@gmail.com (A.B.); tonyziakas@hotmail.com (A.Z.); ggiannakoulas@gmail.com (G.G.); 2Adult Congenital Heart Centre and National Centre for Pulmonary Hypertension, Royal Brompton and Harefield Hospitals, Guy’s and St Thomas’s NHS Foundation Trust, Imperial College, London SW3 5NP, UK; 3Third Department of Obstetrics and Gynecology, School of Medicine, Faculty of Health Sciences, Aristotle University of Thessaloniki, 54642 Thessaloniki, Greece; iotsakir@gmail.com (I.T.); amamop@auth.gr (A.M.); 4Pediatric Department, AHEPA University General Hospital, School of Medicine, Aristotle University of Thessaloniki, 54636 Thessaloniki, Greece; agianop@auth.gr

**Keywords:** d-transposition of the great arteries, congenitally corrected transposition of the great arteries, pregnancy, outcomes, maternal, fetal

## Abstract

**Background:** Pregnancy in women with biventricular circulation and a systemic right ventricle (sRV) is considered high risk, with limited data available on pregnancy outcomes. This study aimed to investigate pregnancy outcomes in this population. **Materials and Methods:** A systematic review was conducted using four major electronic databases. Pregnant women with a complete transposition of great arteries (d-TGA) after an atrial switch operation or a congenitally corrected transposition of the great arteries (ccTGA) were included. **Results:** In total, 15 studies including 632 pregnancies in 415 women with an sRV and biventricular circulation were identified, of whom 299 (72%) had d-TGA and 116 (28%) ccTGA. Maternal mortality or cardiac transplantation occurred in 0.8% of pregnancies. The most frequent maternal complications were the worsening of systemic atrioventricular valve regurgitation [pooled estimate (PE): 16%, 95% CI: 5;26], the deterioration of sRV function (PE: 15%, 95% CI: 2;27), the worsening of the NYHA class (PE: 13%, 95% CI: 6;20), all-cause hospitalization (PE): 10%, 95% CI: 7;12), arrhythmias (PE: 8%, 95% CI: 5;11), and symptomatic heart failure (PE: 6%, 95% CI: 3;10). Stillbirth occurred in 0.7% of pregnancies and neonatal death in 0.4%. Small-for-gestational-age neonates were encountered in 36% (95% CI: 21;52) of pregnancies and preterm delivery in 22% (95% CI: 14;30). A subgroup analysis showed no significant difference in outcomes between women with d-TGA and those with ccTGA, except for the worsening of the NYHA class, which occurred more often in d-TGA (18%, 95% CI: 12;27 vs. 6%, 95% CI: 3;15, respectively, *p* = 0.03). **Conclusions:** Maternal and fetal/neonatal mortality are low among pregnant women with biventricular circulation and an sRV. However, significant maternal morbidity and poor neonatal outcomes are frequently encountered, rendering management in specialized centers imperative.

## 1. Introduction

Nowadays, we are living in the era when patients with d-transposition of the great arteries (d-TGA) who underwent an atrial switch are reaching reproductive age, while women with congenitally corrected transposition of the great arteries (ccTGA) are blessed by advances in surgical techniques and medical management, and they frequently pursue pregnancy [1,2]. Healthcare systems globally are mandated to support the increasing number of women with a systemic right ventricle (sRV) who require specialized multidisciplinary care in expert centers during pregnancy, as they are at high risk for adverse outcomes [3,4].

According to the 2018 European Society of Cardiology (ESC) Guidelines on the management of cardiovascular diseases during pregnancy, women with an sRV and a preserved or mildly reduced ventricular function are considered at high risk to contemplate pregnancy and should be followed-up frequently at tertiary expert centers [5]. Vaginal delivery should be preferred in asymptomatic women with good or moderately reduced RV function, unless worsening of the sRV function is observed during pregnancy. In such cases, an early cesarean section should be scheduled due to high risk of heart failure. On the other hand, women with an sRV and impaired ventricular function are considered at very high risk for adverse maternal and fetal outcomes and therefore pregnancy should be discouraged.

So far, only limited “real-world” data exist regarding maternal and fetal/neonatal complications that may arise during pregnancy in this group of patients. The aim of this systematic review was to summarize and analyze all available data regarding maternal and fetal outcomes during pregnancy in women with a biventricular circulation and an sRV.

## 2. Materials and Methods

This systematic review was conducted adhering to the Preferred Reporting Items for Systematic Reviews and Meta-Analyses (PRISMA) guidelines [6]. The prespecified protocol was submitted in PROSPERO (CRD42024523347).

### 2.1. Search Strategy

A systematic literature review was conducted in four major electronic databases, namely MEDLINE (via PubMed), CENTRAL (Cochrane Central Register of Controlled Trials), Scopus, and Web of Science from inception until November 2023. Article references and gray literature were also searched. PROSPERO was searched for similar systematic reviews in progress to avoid duplication. The search strategy used in PubMed (MESH terms included) and the Cochrane database are presented in the Appendix A. The search was conducted by two independent investigators (T.G.—MD and A.A.—MD).

### 2.2. Eligibility Criteria

We included studies which examined adult pregnant women (≥18 years) with biventricular circulation and an sRV, including those with dextro-looped transposition of the great arteries (d-TGA) after an atrial switch procedure and those with congenitally corrected transposition of the great arteries (ccTGA). Observational original research studies, as well as prospective, retrospective, and case–control ones, were eligible for this systematic review. There were no restrictions concerning geographical location or year of publication. Studies that reported any of the outcomes of interest with at least 10 participants were considered eligible. Case series and case reports were excluded.

### 2.3. Study Outcomes

Maternal outcomes that occurred during pregnancy and up to 6 months after delivery included all-cause mortality, all-cause hospitalization, worsening of the New York Heart Association (NYHA) class, worsening of sRV function, and worsening of systemic atrioventricular valve regurgitation (SAVVR) assessed by echocardiography, any arrhythmic event, symptomatic heart failure, thromboembolic events, endocarditis, newly diagnosed pulmonary arterial hypertension, pre-eclampsia, and gestational arterial hypertension. Moreover, fetal or neonatal outcomes included were spontaneous abortion (<20 weeks of gestation), stillbirth defined as the delivery of a fetus over 20 weeks of gestation without signs of life [7], preterm delivery (birth within 20^+0^ and 36^+6^ weeks) [8], neonatal mortality defined as death within the first month of life, small-for-gestational-age (SGA) neonates (birthweight <10th centile), and those diagnosed with CHD either antenatally or postnatally. The type of delivery (vaginal or cesarean delivery) was also recorded.

### 2.4. Study Selection and Data Collection

All retrieved records were imported into reference management software (Mendeley V.1.19.8/Rayyan). After the removal of duplicates, two independent reviewers (T.G. and A.A) screened the records for eligibility based on the title and abstract level. The full text of the selected studies was assessed for eligibility by these two reviewers, while any discrepancies were resolved by a third reviewer (A.B.—MD).

A structured pilot-tested Excel spreadsheet, designed according to the Cochrane’s Checklist of items [9], was used to extract the information on study design, the number of participants, number of pregnancies, follow-up duration, patients’ baseline characteristics, and outcomes of interest from the full text, figures, tables, and online supplemental appendices. The process was conducted by two independent reviewers (T.G.—MD and A.A.—MD), and discrepancies were solved by a third reviewer (A.B.). Data described with median values and IQRs were converted to mean values and SDs [10]. Missing mean values and SDs were manually calculated from relevant statistics if possible and were otherwise imputed according to Cochrane recommendations [11].

### 2.5. Quality and Publication Bias Assessment

All included studies were critically appraised in terms of quality by two reviewers (T.G. and A.A.) according to the Newcastle–Ottawa Scale [12], while discrepancies were determined by a third reviewer (A.B.). The quality of each study was rated in one of the following categories: “Low Risk”, “Intermediate Risk”, or “High Risk”. Furthermore, publication bias was assessed with a visual inspection of funnel plot asymmetry [13,14]. Finally, in order to determine the overall strength of evidence in our study, the GRADE approach was used by evaluating the risk of bias, inconsistency, indirectness, imprecision, and publication bias.

### 2.6. Statistical Analysis

Extracted data and descriptive statistics, including baseline characteristics and outcomes of interest, are interpreted in Table 1, with missing values presented as NA (not applicable). Studies were evaluated for their heterogeneity with the chi-square test, the random effects model, and the I2 statistic. Sub-group analysis was conducted in order to clarify the cause of heterogeneity. The two groups were defined in terms of the type of TGA (d-TGA or ccTGA) of pregnant women. In case of substantial heterogeneity (I^2^ > 50%), the robustness of our results was explored performing sensitivity analysis with a leave-one-out analysis. The effect estimate of each study was compared against the overall estimate on every outcome [9]. Outcomes were described with the number of events, proportions, pooled estimates (PEs) and 95% confidence intervals (95% CIs). The statistical significance threshold was set at an α-value of 0.05. R statistical software (V.4.2.3) was used for all statistical analyses.

## 3. Results

### 3.1. Search Results

In total, 1941 records were identified in the initial search of the online databases. After excluding duplicates, a total of 1592 records were screened according to their title and abstract, leading to 47 publications selected for full-text analysis. Of these, 33 were excluded for certain reasons (Appendix A). Finally, a total of 15 observational studies fulfilled the eligibility criteria and were included in this systematic review for analysis; 14 studies were retrospective, and one was prospective. (Appendix A).

### 3.2. Baseline Characteristics

The general characteristics of the individual studies are depicted in Table 1. All relevant studies were published in a 15-year interval (2004 and 2022). The median duration of the recruitment of pregnancies among women with biventricular circulation and an sRV was 15 years (IQR 6.75). A total of 632 pregnancies among 415 women with biventricular circulation and an sRV (72% with a dextro-TGA and atrial switch and 28% with a ccTGA) have been reported. Additional congenital heart defects were reported in 43.2% of the participants (*n* = 57/132), including ventricular septal defect (35.7%), atrial septal defect (11.7%), ventricular septal aneurysm (2.4%), patent foramen ovale (2.4%), pulmonary stenosis (26.2%), pulmonary atresia (2.4%), truncus arteriosus (2.4%), Ebstein anomaly of the tricuspid valve (4.6%) and bicuspid aortic valve (2.4%), single coronary artery (2.4%), and dextrocardia (7.1%). The type of surgery was reported in half of d-TGA women (*n* = 151, 50.5%); 56.3% (*n* = 85) of them underwent a Mustard operation and the remaining a Senning procedure. Data regarding an additional cardiac operation performed before pregnancy were available in 37.6% (*n* = 156) of women; 20.5% (*n* = 32) underwent at least one additional procedure (Table 2).

The NYHA class was presented differently among the included studies. Some studies reported the NYHA class per pregnancy [19,20,28], while others per patient. The NYHA classification before pregnancy was reported for 318 patients and for 196 pregnancies. The majority of patients (97.4% pregnancies in 89.6% women) were asymptomatic (NYHA I), 10.1% of patients (2.6% pregnancies) were mildly symptomatic (NYHA II), while only one woman (0.3%) was severely symptomatic (NYHA III) during pregnancy. No woman was reported as NYHA IV. Of 346 women (83.4%) with a history of arrhythmia and/or pacemaker implantation before pregnancy, one quarter (*n* = 88) presented with history of arrhythmias, including atrial tachycardia (15.7%), supraventricular tachycardia (21.6%), ventricular tachycardia (13.7%), atrial fibrillation (17.6%), atrial flutter (3.9%), sick sinus syndrome (3.9%), intermittent nodal rhythm (3.9%), ventricular ectopics (2.2%), intra-atrial reentrant tachycardia (2.2%), and junctional bradycardia (2.2%). A pacemaker was implanted in 15.6% (*n* = 54/346) of women. Echocardiographic data obtained before pregnancy were available in 81% (*n* = 336) of women, of which more than one third (36.6%, *n* = 123) had at least mild sRV dysfunction or SAVVR (36.5%, *n* = 116).

The follow-up duration was between 6 months and 19 years. Three studies did not report follow-up [17,27,28], whereas the exact period was reported in seven studies [16,19,20,22,23,24,29].

### 3.3. Maternal Outcomes

Maternal mortality or cardiac transplantation was reported in 0.8% of pregnancies (*n* = 5); one death occurred at the 27th week of gestation, and another three women died at 6 weeks and 3 and 6 months postpartum, while one woman underwent cardiac transplantation during the postpartum period. Three women had a d-TGA with an atrial switch operation and two had a ccTGA (Table 3).

Maternal complications are presented in Figure 1 and Table 4. The most common maternal complications were the worsening of SAVVR presented in 16% (*n* = 71/490) of pregnancies (95% CI: 5;26, I2 = 87%, *p* value < 0.001) and the worsening of sRV function in 15% (*n* = 37/256) of them (CI: 2;27, I2 = 86%, *p* value < 0.001). Deterioration of the NYHA class presented in 13% (*n* = 33/216) of pregnancies (95% CI: 6;20, I2 = 57%, *p* value = 0.02) and persisted postpartum in 19 out of 30 pregnancies (63.3%) for which data were available [16,17,19,22,24,25,27,29]. Hospitalization due to any cause was required in 10% (*n* = 44/384) of pregnancies (95% CI: 7;12, I2 = 40%, *p* value = 0.11); the main reason for admission was cardiac causes in 77.3% of patients, while 22.7% hospitalizations were for obstetric [29] or undefined reasons [2]. The main cardiac causes for hospitalization were symptomatic heart failure, deterioration of the NYHA, and the presence of arrhythmia [17,19,21,23,27,28].

Arrhythmias throughout gestation occurred in 8% (*n* = 51/534) of pregnancies (95% CI: 5;11, I2 = 40%, *p* value = 0.07). The most common types of arrhythmias were supraventricular tachycardia (21.6%), atrial fibrillation (17.6%), atrial tachycardia (15.7%), and ventricular tachycardia (13.7%), while other types occurred in 3.9% of pregnancies (Table 5).

Symptomatic heart failure was diagnosed in 52 of 594 pregnancies (PE: 6%, 95% CI: 3;10, I2 = 62%, *p* value < 0.01). The diagnosis was attributed to echocardiographic measurements.

Other inconsistently reported complications were a new baffle leak or obstruction reported in 13 pregnancies, preeclampsia (11 pregnancies), gestational hypertension (9 pregnancies), while thromboembolic events, including transient ischemic attack (TIA), stroke, and pulmonary embolism, were reported in seven pregnancies. Less frequent complications were endocarditis (three pregnancies) and pulmonary arterial hypertension (PAH), myocardial infarction, toxemia, and hemoptysis, complicating one pregnancy each.

### 3.4. Fetal/Neonatal Outcomes

From the 632 pregnancies, data regarding fetal mortality were available in 567 of them. In total, fetal mortality was reported in 7.9% of pregnancies (*n* = 45), with four stillbirths (0.7%) (Table 6) and 41 spontaneous abortions (PE 6%, 95% CI: 2;10, I2 = 75%, *p* value < 0.01). In addition, 21 induced abortions during the first trimester (PE 2%, 95% CI: 0;4, I2 = 47%, *p* value = 0.03) were recorded.

The mean pregnancy duration was estimated as 36.6 weeks (SD, 0.88). Vaginal delivery was opted for in half of the pregnancies (51%, 95% CI: 37;65, I2 = 94%, *p* value < 0.01) and cesarean section in more than one-third (35%, 95% CI: 21;49, I2 = 93%, *p* value < 0.01), while missing data were recorded in 14% of pregnancies (*n* = 89/632) [16,21]. Preterm delivery occurred in 22% (95% CI: 14;30, I2 = 89%, *p* value < 0.01) of 570 pregnancies.

As for neonatal outcomes, the mean birthweight was 2723.07 g (SD 188.4). SGA neonates were reported in 128 out of 431 live births with available data with a PE of 36% (95% CI: 21;52, I2 = 92%, *p* value < 0.01). Moreover, three (0.4%) neonates died out of 622 pregnancies with available data regarding neonatal mortality, with two due to cerebral palsy and one due to sepsis. All of them were born prematurely (<37 weeks). CHD was diagnosed in three neonates, but the type of CHD was not reported.

Fetal and neonatal outcomes are presented in Figure 2 and Table 7.

### 3.5. Comparison Between d-TGA After Atrial Switch Operation and ccTGA

A subgroup analysis between pregnant women with d-TGA after an atrial switch operation and those with ccTGA was conducted. The incidence of worsening of the NYHA class was significantly higher in the d-TGA group (PE: 6%, 95% CI: 3;15 vs. 18%, 95% CI: 12;27, respectively, *p* = 0.03) compared to the cc-TGA. No statistical difference was observed in worsening of the sRV function (PE: 16%, 95% CI: 3;53 vs. 9%, 95% CI: 3;26, respectively), the need for hospitalization during pregnancy (PE: 9%, 95% CI: 5;15 vs. 18%, 95% CI: 6;42, respectively), and prematurity (PE: 16%, 95% CI: 9;26 vs. 28%, 95% CI: 19;38, respectively) between patients with cc-TGA and d-TGA (Figure 3).

### 3.6. Sensitivity Analysis

Apart from the subgroup analysis, we tried to elucidate the source of heterogeneity further with sensitivity analysis by performing a leave-one-out analysis. Statistical heterogeneity among individual studies was obtained using Cochrane’s Q test and the I^2^ statistic; a value >50% indicated substantial heterogeneity, so a sensitivity analysis was conducted. After removing the outlier studies, the I^2^ did not decrease under 50%, suggesting that these studies did not alter the overall conclusions and were not the only ones that contributed to heterogeneity.

### 3.7. Quality Assessment and Publication Bias of Included Studies

The quality assessment of the included studies is reported in the Appendix A. Due to the prespecified design of the included studies, none of them were adjusted for potential cofounders. Six studies were rated as six stars—moderate quality—and nine of them as five stars—moderate quality—due to lack of comparability. Furthermore, evidence of publication bias was observed across all outcomes (*p* < 0.05), with the exception of all-cause hospitalization (t = 5.79, *p*-value = 0.0012). (Appendix A).

The overall rating of the quality of evidence was assessed by the GRADE approach. Since the included studies were observational, the initial evaluation started at low certainty. The quality was downgraded for three elements, including potential risk of bias, inconsistency due to substantial heterogeneity, and potential publication bias. However, certainty was upgraded for illustrating a dose–response relationship, as the results were proportional to the degree of outcomes. The certainty of evidence was rated as low, driven by study limitations, substantial heterogeneity, and potential publication bias.

## 4. Discussion

This systematic review offers detailed insights into maternal and fetal/neonatal complications in pregnant women with an sRV and biventricular circulation. Overall, maternal and fetal mortality rates were low, and pregnancy was well tolerated in the majority of these women, with a low incidence of major cardiac events. Sudden cardiac death was the prevalent mode of death, while pre-existing heart failure and arrhythmias were significant contributors for maternal mortality. Cardiac transplantation was required in one case post-partum. The most frequent pregnancy complications were preterm delivery and SGA. Long-term maternal complications could not be adequately addressed due to the short post-partum follow-up period of the included studies.

### 4.1. Maternal Outcomes

The majority of women included in this study were asymptomatic or mildly symptomatic at baseline, with a significant worsening of the NYHA class during pregnancy. This deterioration could be attributed to the fact that patients with an sRV fail to sufficiently increase cardiac output as a physiological response to increased metabolic demands during pregnancy. Furthermore, the deterioration of sRV function and the worsening of systemic atrioventricular valve regurgitation that may occur during pregnancy could explain symptomatic deterioration [17,30,31].

Symptomatic heart failure was encountered in 6% of pregnancies and may be attributed to sustained hemodynamic changes, including the increase in blood volume, stroke volume, and cardiac output during the second and third trimester [28,29,30,32,33]. Diagnosis of heart failure during pregnancy should not be based on clinical evaluation, as dyspnea and fatigue may also present in uncomplicated pregnancies [34,35,36]. Therefore, serial echocardiographic and biomarker assessment should be considered during pregnancy in these patients [5]. The time of onset of heart failure in our study varied between the second and third trimester, where several risk factors, including myocardial fibrosis, ischemia and perfusion defects, systemic atrioventricular valve regurgitation, and arrhythmias may have played a role [36].

The risk of maternal arrhythmia was 8%, with the most common type being supraventricular tachycardia, which could be attributed to a combination of hormonal influence, structural and electrical abnormalities, and pregnancy-induced hemodynamic changes that led to atrial enlargement and stretch [37]. Nevertheless, the time of primal arrhythmia could not be determined. None of the patients required the administration of new medications during pregnancy. While angiotensin-converting enzyme (ACE) inhibitors, angiotensin receptor blockers (ARBs), and amiodarone were discontinued during pregnancy, the use of b-blockers, digoxin, aspirin, furosemide, and methyldopa was continued as indicated [2,17,19,20,21,25,26,28,29]. According to previous studies, the incidence of arrhythmia in pregnant women with CHD varied between 0.9 and 12.7% in the most complex lesions [38].

Regarding maternal mortality or the cardiac transplantation rate, 0.8% of pregnant women with an sRV and biventricular circulation died during pregnancy or the puerperium. The three studies that reported maternal mortality or the need for cardiac transplantation were published 18, 12, and 4 years ago, and patients were recruited between 1985 and 2018 [18,21,27]. As a result, preconception counseling, pregnancy planning, and overall management for these patients may have varied from current guideline-directed practices. The results of our study are in accordance with the ROPAC registry of pregnancy and cardiac disease, where the total maternal mortality rate was 0.6% among pregnant women with any type of cardiovascular disease including valvular, ischemic heart disease, cardiomyopathy, aortopathy, pulmonary arterial hypertension, and CHD. The highest rates were reported in the PAH (9%), cardiomyopathy (1.1%), and valvular heart disease (1%) groups. Mortality rate in the CHD group was 0.2%; however, large heterogeneity may have affected the result [39].

When comparing pregnancy outcomes in patients with an sRV with those in women with d-TGA after an arterial switch operation, the rate of maternal complications is not remarkably higher. Hardee et al. reported the incidence of mortality to be only 1% higher in the sRV group. In addition, major cardiac events, including arrhythmia and symptomatic heart failure, did not differ significantly [35].

### 4.2. Fetal/Neonatal Outcomes

The most common fetal complications observed were SGA neonates (36%) and prematurity (22%). Notably, the risk was higher than that reported by Pizula et al. for pregnancies of women with d-TGA after an arterial switch procedure (*n* = 120 pregnancies), in which SGA and premature neonates were 9% and 11%, respectively [40]. SGA and prematurity could be attributed to low cardiac output and the inability of the heart to support the hyperdynamic circulation during pregnancy [16,17,24], while another theory could associate these outcomes with the maternal use of cardiac medication during pregnancy (b-blockers, antiplatelet therapy, diuretics) [28,41]. However, the 2018 ESC guidelines for the management of cardiovascular disease during pregnancy recommend that b-blockers should be continued during pregnancy and selective beta-1 blockers should be preferred, while nonselective b-blockers have been associated with fetal growth restriction (FGR) [42]. With regard to prematurity, it may be attributed to iatrogenic preterm delivery to minimize the maternal risks during pregnancy. Stillbirth and neonatal mortality rates were relatively low in our study (0.7% and 0.4%, respectively); they were lower rates than the ones reported from the ROPAC registry among pregnant women with any type of cardiovascular disease (1.3% and 0.6%, respectively) [39].

Notably, vaginal delivery was preferred among patients with an sRV, while the PE rate of cesarean section was 35%. However, in four studies, the proportion of cesarean section was remarkably higher than the average, at 51.6%, 45.2%, 83.3%, and 48.8%, respectively [22,25,28,29]. The reason for such a high rate of cesareans was not defined in three studies. Only Tutarel et al. reported the indication for a cesarean section; in 32.9% of the cases, a cesarean section was performed due to a cardiac reason, including severe cardiac disease (34%), heart failure (23%), arrhythmia (15%), and the use of anticoagulation therapy (8%).

### 4.3. D-TGA After Atrial Switch Operation Versus ccTGA

A worsening of the NYHA class during pregnancy occurred more often in the d-TGA group than ccTGA (18% vs. 6%, *p* value = 0.03). Although the NYHA classification may be a subjective measure of symptoms, the worsening of the NYHA class reflects a reduction in exercise capacity in pregnant women with d-TGA and an atrial switch operation, which could be associated with chronotropic incompetence, limited tolerance to hemodynamic changes during pregnancy, and susceptibility to heart failure and arrhythmia [33]. On the other hand, women with ccTGA seem to tolerate pregnancy better compared to those with d-TGA. However, the difference in symptom worsening could not be translated into a difference in maternal or fetal outcomes between the two groups.

The ESC-EORP registry of pregnancy and cardiac disease reported that significant variance occurred in hospital admission due to a cardiac reason, which was reported more often in the ccTGA group (19.5% vs. 6.6%, respectively) [39]; nevertheless, in our analysis, a numerically remarkable difference was observed; however, it was not statistically significant.

### 4.4. Study Limitations

Certain limitations should be acknowledged; this systematic review included only observational retrospective cohort studies and therefore, there may be missing data regarding the baseline characteristics of the patients and the pregnancy outcomes. A selection bias may exist, since all pregnant women were managed in expert centers, as recommended by the 2018 ESC Guidelines for high-risk pregnancies [41]. Furthermore, high statistical heterogeneity was observed among the included studies. We tried to elucidate the heterogeneity using the random effects model and performing subgroup and sensitivity analyses. Moreover, most of the pregnant women included were asymptomatic or presented with mild complications; it is probable women with a more severe cardiac disease were counseled against pregnancy or terminated it. Therefore, the results of our study might not be applicable to high-risk pregnant women with biventricular circulation and an sRV. Moreover, the correlation between biventricular circulation with an sRV and the likelihood of spontaneous abortions in our study might be underestimated due to the exclusion of nonviable pregnancies in some studies. Last but not least, postpartum follow-up for adverse maternal outcomes was insufficient and long-term maternal complications could not be adequately addressed.

## 5. Conclusions

In conclusion, pregnancy in asymptomatic or mildly symptomatic women with a biventricular circulation and an sRV can be well tolerated with a low rate of maternal and fetal mortality. Sudden cardiac death may appear; however, heart failure is also a significant contributor to maternal mortality. A number of cardiac events demonstrated low rates of maternal morbidity. However, they should not be ignored, as significant complications may still arise. Individualized pre-conceptional counseling, close evaluation throughout gestation and postpartum at expert centers, and the development of a personalized pregnancy and delivery plan could be the keys for an uneventful peripartum period [5,43] (Appendix B, Figure A1). Further prospective longitudinal multicenter studies in larger populations are encouraged to guide pregnancy counseling, management, and follow-up.

## Figures and Tables

**Figure 1 jcm-13-07281-f001:**
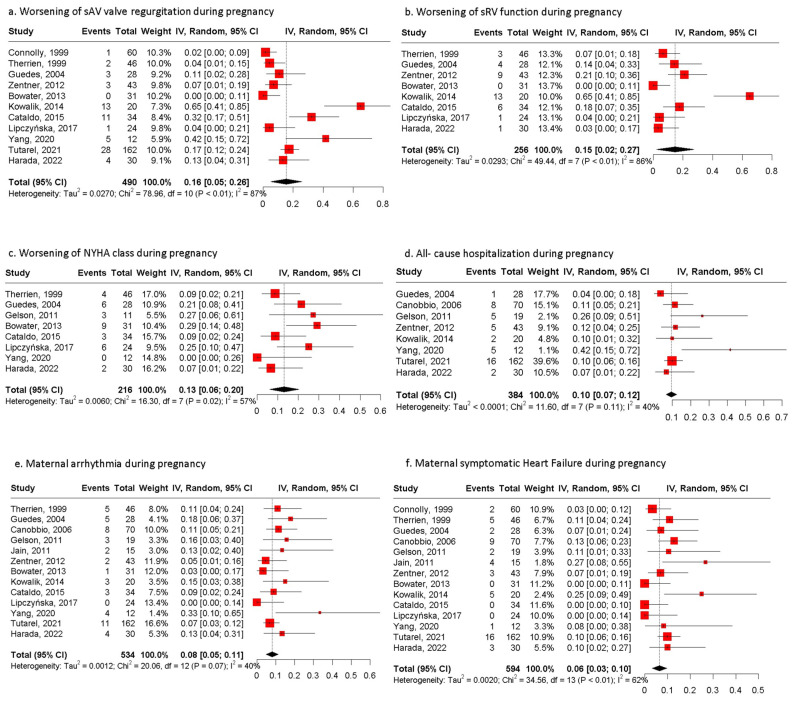
Pooled estimates of maternal outcomes during pregnancy. (**a**) Forest plot of worsening of systemic AV valve regurgitation during pregnancy. (**b**) Forest plot of worsening of systemic RV function during pregnancy. (**c**) Forest plot of worsening of NYHA class during pregnancy. (**d**) Forest plot of maternal all-cause hospitalization during pregnancy. (**e**) Forest plot of maternal arrythmia during pregnancy. (**f**) Forest plot of maternal symptomatic heart failure during pregnancy; NYHA: New York Heart Association [15,16,17,19,20,21,23,24,25,27,28,29,30].

**Figure 2 jcm-13-07281-f002:**
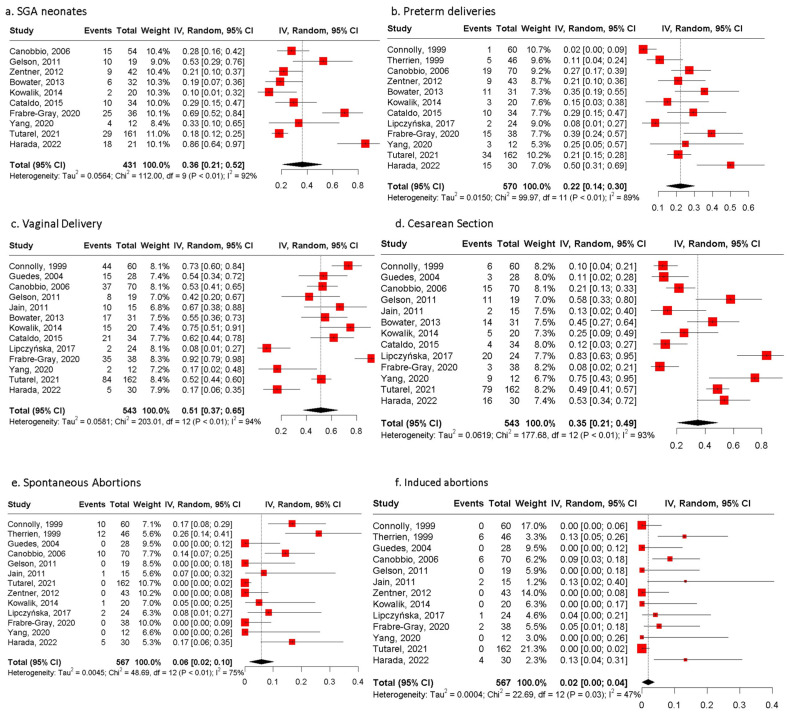
Pooled estimates of fetal/neonatal and pregnancy outcomes of women with biventricular circulation and an sRV. (**a**) Forest plot of SGA neonates. (**b**) Forest plot of preterm deliveries. (**c**) Forest plot of vaginal deliveries. (**d**) Forest plot of cesarean sections. (**e**) Forest plot of spontaneous abortions of women with biventricular circulation and an sRV. (**f**) Forest plot of induced abortions of women with biventricular circulation and an sRV reported. SGA: small for gestational age [15,16,17,18,19,20,21,22,23,24,25,26,27,28,29].

**Figure 3 jcm-13-07281-f003:**
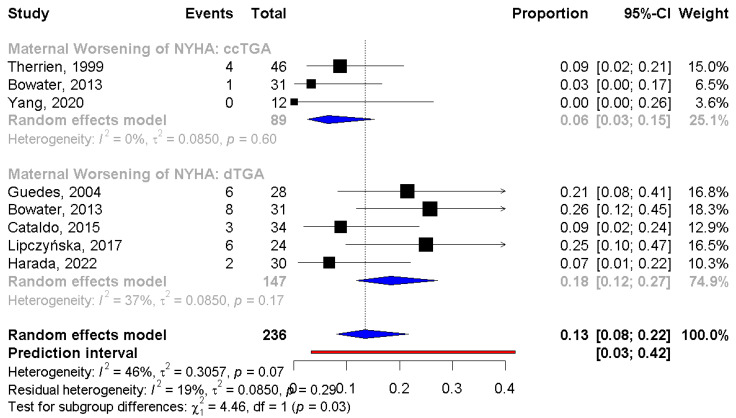
Subgroup analysis for NYHA class between d-TGA and ccTGA group. NYHA: New York Heart Association; d-TGA: dextro-looped transposition of the great arteries; ccTGA: congenitally corrected transposition of the great arteries [16,17,19,22,24,25,27,29].

**Table 1 jcm-13-07281-t001:** Baseline characteristics of pregnant women with biventricular circulation and an sRV; d-TGA: dextro-looped transposition of the great arteries; ccTGA: congenitally corrected transposition of the great arteries; NYHA: New York Heart Association; SV: systemic ventricle; SAV valve: systemic atrioventricular valve; CHD: congenital heart disease; NA: Not Applicable.

Author, Year	Study Design	Country	No of Patients	No of Pregnancies	Age at Pregnancy (Mean ± SD), y	Senning Repair	Mustard Repair	dTGA	ccTGA	NYHA Class/Patients	NYHA Class/Pregnancies	SV Dysfunction	SV Regurgitation *	No of Patients with Additional CHD	No of Patients with Additional Operation Before Pregnancy	History of Arrythmia	Pacemaker
Connolly, 1999 [15]	retrospective	USA	22	60	28.6 ± 5.6	0	0	0	22	NA	NA	NA	NA	10	9	1	1
Therrien, 1999 [16]	retrospective	UK	19	46	27 ± 12.5	0	0	0	19	I-14 II-5	NA	NA	NA	13	7	7	7
Guedes, 2004 [17]	retrospective	Canada	16	28	27 ± 5	0	16	16	0	I-11 II-5	NA	4	4	NA	NA	10	5
Canobbio, 2006 [18]	retrospective	USA	40	70	NA	4	36	40	0	I-31 II-8 III-1	NA	19	22	NA	6	16	13
Gelson, 2011 [19]	retrospective	UK	14	19	29.1 ± 5.7	0	11	11	3	NA	I-16 II-3	6	NA	6	5	7	5
Jain, 2011 [20]	retrospective	USA	10	15	26.7 ± 0.5	NA	NA	8	2	NA	Ι-13 ΙΙ-2	4	NA	NA	NA	6	6
Zentner, 2012 [21]	retrospective	Australia	19	43	NA	18	1	19	0	NA	NA	NA	NA	2	NA	NA	NA
Bowater, 2013 [22]	retrospective	UK	18	31	25.5 ± 5.2	9	5	14	4	I-18	NA	8	27	NA	NA	NA	NA
Kowalik, 2014 [23]	retrospective	Poland	13	20	26.6 ± 5.6	0	0	0	13	NA	NA	13	NA	7	0	NA	0
Cataldo, 2015 [24]	retrospective	UK	21	34	26 ± 6	10	11	21	0	I-18 II-3	NA	6	3	NA	3	7	1
Lipczyńska, 2017 [25]	retrospective	Poland	15	24	24.8 ± 3.2	14	1	15	0	I-14 II-1	NA	0	1	1	NA	6	2
Frabre-Gray, 2020 [26]	retrospective	UK	19	38	24.5 ± 2.5	NA	NA	19	0	NA	NA	NA	3	9	NA	NA	NA
Yang, 2020 [27]	retrospective	China	12	12	27.1 ± 3.7	0	0	0	12	I-9 II-3	NA	11	9	9	1	4	0
Tutarel, 2021 [28]	prospective	international	162	162	28.8 ± 4.6	NA	NA	121	41	I-162	I-162	44	44	NA	NA	14	14
Harada, 2022 [29]	retrospective	Tokyo	15	30	29 ± 9	11	4	15	0	I-8 II-7	NA	8	3	NA	1	10	0
			415	632	27 ± 1.5	66 (43,7)	85 (56.3)	299 (72)	116 (28)	I- 285 (89.6) II- 32 (10.1) III- 1 (0.3) IV- 0	I- 191 (97.4) II- 5 (2.6) III- 0 IV- 0	123 (36.6)	67 (36.5)	57 (43.2)	32 (20.51)	88 (25.4)	54 (15.6)

* indicates at least moderate SV regurgitation.

**Table 2 jcm-13-07281-t002:** Additional operations performed before pregnancy.

Palliative Operation	No of Events
VSD closure	7
ASD closure	3
LV to PA conduit	7
PA banding	2
Pulmonary valvotomy	1
Systemic atrioventricular valve replacement	3
Tricuspid valve valvuloplasty	3
Surgical baffle revision	5
Percutaneous intervention for venous pathway obstruction.	4
Fontan circulation	1
Blalock–Taussing shunt placement	1

VSD: ventricular septal defect; ASD: atrial septal defect; LV: left ventricle; PA: pulmonary artery.

**Table 3 jcm-13-07281-t003:** Maternal mortality or cardiac transplantation cases.

Author, Year	Maternal Mortality	Details
Canobbio, 2006 [18]	2	dTGA, Case 1: Sudden cardiac death in a woman with an sRV and atrial switch procedure occurred at 6 weeks postpartum after emergency delivery <34 weeks of gestation. She developed heart failure on the third trimester of pregnancy.
		dTGA, Case 2: Severe right ventricular heart failure occurred in a woman with an sRV and atrial switch procedure during the third trimester of pregnancy and led to cardiac transplantation after labor.
Zentner, 2012 [21]	2	dTGA, Case 1: In a woman with an sRV after an atrial switch procedure, sudden cardiac death occurred at 27 weeks of gestational age. Three weeks before, palpitations with a loss of consciousness were reported and cardiac death was imputed to arrhythmia.
		dTGA, Case 2: Sudden cardiac death occurred in a woman six months postpartum. During pregnancy, atrial arrhythmias and heart failure were reported. Both beta blocker (sotalol) and ACE inhibitor (enalapril) were terminated before and during pregnancy and were recommended during the postpartum period.
Yang, 2020 [27]	1	ccTGA. In a 23-year-old woman with situs solitus, ccTGA, VSD, and PS, cardiac death occurred due to ventricular fibrillation 3 months postpartum. At 36 weeks gestational age, she presented to the emergency department with sinus tachycardia.
Total Cases	5	

d-TGA: dextro-looped transposition of the great arteries; sRV: systemic right ventricle; ACE inhibitor: angiotensin-converting enzyme inhibitors; ccTGA: congenitally corrected transposition of the great arteries; VSD: ventricular septal defect; PS: pulmonary stenosis.

**Table 4 jcm-13-07281-t004:** Maternal outcomes of pregnant women with biventricular circulation and an sRV. NYHA: New York Heart Association; SV: systemic valve; TIA: transient ischemic attack; NA: Not Applicable.

Author, Year	Maternal Mortality	Arrhythmia	Symptomatic Heart Failure	Worsening of NYHA	Worsening of SV Dysfunction	Worsening of SV Regurgitation	Thromboembolic Events/TIA/Stroke	Baffle Complications	Preeclampsia	All-Cause Hospitalization
Connolly, 1999 [15]	0	NA	2	NA	NA	1	NA	NA	NA	NA
Therrien, 1999 [16]	0	5	5	4	3	2	NA	NA	NA	NA
Guedes, 2004 [17]	0	5	2	6	4	3	0	NA	NA	1
Canobbio, 2006 [18]	1	8	9	NA	NA	NA	NA	NA	4	8
Gelson, 2011 [19]	0	3	2	3	NA	NA	NA	NA	1	5
Jain, 2011 [20]	0	2	4	NA	NA	NA	0	NA	NA	NA
Zentner, 2012 [21]	2	2	3	NA	9	3	ΝA	NA	1	5
Bowater, 2013 [22]	0	1	0	9	0	0	0	0	NA	NA
Kowalik, 2014 [23]	1	3	5	NA	13	13	NA	NA	0	2
Cataldo, 2015 [24]	0	3	0	3	6	11	2	5	NA	NA
Lipczyńska, 2017 [25]	0	0	0	6	1	1	0	0	0	NA
Frabre-Gray, 2020 [26]	0	NA	NA	NA	NA	NA	2	4	1	NA
Yang, 2020 [27]	1	4	1	0	NA	5	NA	NA	1	5
Tutarel, 2021 [28]	0	11	16	NA	NA	28	3	NA	3	16
Harada, 2022 [29]	0	4	3	2	1	4	NA	4	NA	2
TOTAL	5 (0.8)	51 (8)	52 (6)	33 (13)	37 (15)	71 (16)	7	13	11	44

**Table 5 jcm-13-07281-t005:** Types of cardiac arrhythmias occurring during pregnancy in women with biventricular circulation and an sRV. SSS: sick sinus syndrome; IART: intra-atrial re-entrant tachycardia; NA: Not Applicable.

Author, Year	No of Arrhythmia	Atrial Tachycardia	Atrial Fibrillation	Sick Sinus Syndrome (SSS)	Atrial Flutter	Supraventricular Tachycardia	Ventricular Tachycardia	Ventricular Ectopics	IART	Juctional Bradycardia	Intermittent Nodal Rhythm
Connolly, 1999 [15]	NA	NA	NA	NA	NA	NA	NA	NA	NA	NA	NA
Therrien, 1999 [16]	5	NA	1	NA	NA	NA	NA	NA	NA	NA	NA
Guedes, 2004 [17]	5	NA	NA	NA	NA	4	NA	NA	NA	1	NA
Canobbio, 2006 [18]	8	NA	7	NA	1	NA	NA	NA	NA	NA	NA
Gelson, 2011 [19]	3	2	NA	NA	NA	NA	NA	1	NA	NA	NA
Jain, 2011 [20]	2	NA	NA	NA	NA	NA	NA	NA	NA	NA	NA
Zentner, 2012 [21]	2	1	NA	NA	NA	NA	NA	NA	1	NA	NA
Bowater, 2013 [22]	1	NA	NA	NA	1	NA	NA	NA	NA	NA	NA
Kowalik, 2014 [23]	3	NA	NA	NA	NA	2	NA	NA	NA	NA	NA
Cataldo, 2015 [24]	3	2	1	NA	NA	NA	NA	NA	NA	NA	NA
Lipczyńska, 2017 [25]	0	NA	NA	NA	NA	NA	NA	NA	NA	NA	NA
Frabre-Gray, 2020 [26]	NA	NA	NA	NA	NA	NA	NA	NA	NA	NA	NA
Yang, 2020 [27]	4	1	NA	NA	NA	NA	1	NA	NA	NA	2
Harada, 2022 [29]	4	2	NA	2	NA	NA	NA	NA	NA	NA	NA
Tutarel, 2021 [28]	11	NA	NA	NA	NA	5	6	NA	NA	NA	NA
TOTAL	51	8 (15.7)	9 (17.6)	2 (3.9)	2 (3.9)	11 (21.6)	7 (13.7)	1 (2)	1 (2)	1 (2)	2 (3.9)

**Table 6 jcm-13-07281-t006:** Stillbirths.

Author, Year	No. of Stillbirths (*n*)	Details
Connolly, 1999 [15]	1	No further information was provided.
Tutarel, 2021 [28]	1	No further information was provided.
Zentner, 2012 [21]	2	Unexplained death. A fetus died during the 25th week of gestation.

**Table 7 jcm-13-07281-t007:** Fetal/neonatal outcomes. SGA: small for gestational age; APGAR score: Appearance, Pulse, Grimace, Activity and Respiration score; CHD: congenital heart disease; NA: Not Applicable.

Author, Year	Live Births	Stillbirth	Neonatal Mortality	Spontaneous Abortions	Elective Abortions	Induced Labor	Vaginal Labor	Cesarean Section	Prematurity	Pregnancy Duration	SGA	Average Birth Weight	Low APGAR Score	CHD
Connolly, 1999 [15]	50	1	0	10	0	NA	44	6	1	NA	NA	3200 ± 400	NA	0
Therrien, 1999 [16]	27	0	0	12	6	NA	NA	NA	5	NA	NA	NA	NA	1
Guedes, 2004 [17]	18	0	0	0	0	NA	15	3	NA	38.1 ±1.5	NA	3.040 ± 540	NA	0
Canobbio, 2006 [18]	54	0	0	10	6	NA	37	15	19	36.7 ± 3.6	15	2.714 ± 709	NA	NA
Gelson, 2011 [19]	19	0	0	0	0	NA	8	11	NA	37.5	10	2633	NA	NA
Jain, 2011 [20]	15	0	0	1	2	8	10	2	NA	36 ± 2.3	NA	2621.7 ± 514.3	NA	1
Zentner, 2012 [21]	42	2	0	0	0	NA	NA	NA	9	35 ± 1.25	9	2700 ± 275	NA	0
Bowater, 2013 [22]	32	0	1	NA	NA	NA	17	14	11	(30–41)	6	NA	NA	0
Kowalik, 2014 [23]	20	0	0	1	0	NA	15	5	3	NA	2	NA	NA	1
Cataldo, 2015 [24]	34	0	0	NA	NA	12	21	4	10	37 ± 2.5	10	2525 ± 517.5	NA	0
Lipczyńska, 2017 [25]	22	0	0	2	1	NA	2	20	2	37.2 ± 3.75	NA	NA	NA	0
Frabre-Gray, 2020 [26]	36	0	2	0	2	NA	35	3	15	37 ± 2	25	2515 ± 517.5	5	0
Yang, 2020 [27]	12	0	0	0	0	NA	2	9	3	36.5 ± 1.25	4	3010 ± 942.8	0	0
Tutarel, 2021 [28]	161	1	0	0	0	NA	84	79	34	NA	29	NA	12	NA
Harada, 2022 [29]	21	0	0	5	4	NA	5	16	15	36 ± 2.5	18	2272 ± 606.75	NA	NA
TOTAL	563 (84.1)	4	3,0	41 (6)	21 (2)	20.0	295 (51)	187 (35)	127 (22)	36,6 ± 0,88	128 (36)	2723.07 ± 188.4	17	3.0

## Data Availability

All data relevant to the study are included in the article or uploaded in the Appendix A.

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
