# Peer review of "Pregnancy Outcomes in Women with Biventricular Circulation and a Systemic Right Ventricle: A Systematic Review"

_jcm, 2024, doi:10.3390/jcm13237281_

Round 1
Reviewer 1 Report
Comments and Suggestions for Authors
This is a very interesting article about the maternal and fetal outcomes among pregnant women with biventricular circulation and systemic right ventricle.
The methodology of this systematic review and meta-analysis is well conducted, however, there are some weakness point:
-firstly, authors didn’t declare the MESH terms used in the literature search
-a flow chart diagram demonstrating inclusion and esclusion criteria in the final selection of articles is missing
-Table 3 and 6 are not well depicted; they could be more impressive
-there is only the forest plot about subgroup analysis for NYHA class between d-TGA and ccTGA group; while authors declared to not observe no statistical difference regarding worsening sRV function, need for hospitalization, prematurity but none of these was reported in the Figure 3, as previously mentioned
Comments on the Quality of English LanguageOverall, English level is good.
Author Response
|
Dear Sir/Madame,
Thank you very much for taking the time to review this manuscript. We believe that our manuscript has been substantially improved following a thorough revision based on reviewer’s thoughtful comments and we hope to be fit for publication in Journal of Clinical Medicine. Please find the detailed responses below. |
|
|
Reviewer 1: The methodology of this systematic review and meta-analysis is well conducted, however, there are some weakness points:
1. Firstly, authors didn’t declare the MESH terms used in the literature search. Authors’ Response: We thank the reviewer for pointing this out. We have included the literature search, including the MESH terms in the supplementary appendix; appendix S1, page 3-4, PubMed search. We have also made a comment in main manuscript to highlight it as follows:
2. A flow chart diagram demonstrating inclusion and exclusion criteria in the final selection of articles is missing.
Authors’ Response: We thank the reviewer for their comment. A Prisma 2020 flow diagram for new systematic reviews which included searches of databases and registers was included in the supplementary appendix; Figure S1, page 8.
3. Table 3 and 6 are not well depicted; they could be more impressive.
Authors’ Response: We thank the reviewer for their suggestion. All information provided for each case was included. Unfortunately, no further details could be extracted from original studies.
4. There is only the forest plot about subgroup analysis for NYHA class between d-TGA and ccTGA group; while authors declared to not observe no statistical difference regarding worsening sRV function, need for hospitalization, prematurity but none of these was reported in the Figure 3, as previously mentioned.
Authors’ Response: We thank the reviewer for their remark. Forest plots of subgroup analysis of outcomes without a statistically significant difference between the two groups were added on supplementary appendix; Table S4, page 14.
|
|
|
|
|
|
Your sincerely, |
Reviewer 2 Report
Comments and Suggestions for Authors
The manuscript provides a systematic review of pregnancy outcomes in women with biventricular circulation and a systemic right ventricle, focusing on maternal and neonatal complications. It is a valuable topic due to the high-risk nature of pregnancy in this population, where evidence is limited. Although the review is based on observational retrospective studies, which could introduce bias in the results, and reduce the strength of the evidence, the methodology is solid, and the message is clear: “pregnancy in asymptomatic or mildly symptomatic women with a biventricular circulation and sRV can be well tolerated”.
I would like to commend the authors for conducting this thorough and well-structured analysis, following the PRISMA guidelines comprehensively.
Please find below some individual comments:
1) The authors state that “high statistical heterogeneity was observed among the included studies”. Beyond the subgroup analyses, sensitivity analyses (e.g. leave-one-out analyses) could also be conducted to explore the impact of each individual study on the overall results (Maybe a particular study is driving the heterogeneity). This would increase robustness of the results.
2) In the same vein, the authors might consider using the GRADE approach to evaluate the certainty of evidence for each outcome. This would provide a systematic assessment of the quality of the evidence, especially in the context of observed heterogeneity
3) “A worsening of NYHA class during pregnancy occurred more often in the d-TGA
group than ccTGA (18% vs 6%, p value= 0.03)”.
The authors could elaborate more on the clinical implications of the higher incidence of worsening NYHA class during pregnancy in the d-TGA group, in the discussion section.
Author Response
|
Dear Sir/Madame,
Thank you very much for taking the time to review this manuscript. We believe that our manuscript has been substantially improved following a thorough revision based on reviewer’s thoughtful comments and we hope to be fit for publication in Journal of Clinical Medicine. Please find the detailed responses below. |
|
Reviewer 2: 1) The authors state that “high statistical heterogeneity was observed among the included studies”. Beyond the subgroup analyses, sensitivity analyses (e.g. leave-one-out analyses) could also be conducted to explore the impact of each individual study on the overall results (Maybe a particular study is driving the heterogeneity). This would increase robustness of the results.
Authors’ Response: We thank the reviewer for their comment. Sensitivity analysis, using the leave-one-out method, was conducted in order to increase robustness of our results. We have modified manuscript accordingly as follows:
2) In the same vein, the authors might consider using the GRADE approach to evaluate the certainty of evidence for each outcome. This would provide a systematic assessment of the quality of the evidence, especially in the context of observed heterogeneity.
Authors’ Response: We thank the reviewer for their remark. Indeed, GRADE approach was performed to evaluate the overall certainty of evidence. Risk of bias assessment of the included studies was also rated by Newcastle- Ottawa Scale, and publication bias by funnel plot and Egger’s regression test. GRADE approach was updated on pages 4, 14 of the manuscript. Risk of bias and publication bias assessments were included in supplementary appendix; Table S3 & S4, respectively, page 12-13.
3) “A worsening of NYHA class during pregnancy occurred more often in the d-TGA group than ccTGA (18% vs 6%, p value= 0.03)”. The authors could elaborate more on the clinical implications of the higher incidence of worsening NYHA class during pregnancy in the d-TGA group, in the discussion section.
|
||
|
Authors’ Response: We thank the reviewer for their comment. We have added a comment regarding clinical implications of NYHA worsening on discussion section, page 16 of manuscript.
Your sincerely, |